# The Importance of Monitoring the Work-Life Quality during the COVID-19 Restrictions for Sustainable Management in Nursing

Mateja Lorber [1],* and Mojca Dobnik [1,2]

1   Faculty of Health Sciences, University of Maribor, 2000 Maribor, Slovenia
2   Ministry of Health, 1000 Ljubljana, Slovenia
*   Correspondence: mateja.lorber@um.si; Tel.: +386-230-047-02

**Abstract:** The aim of this study was to investigate the work-life quality and related workplace factors of nursing employees working in hospitals during the COVID-19 restrictions. Employees in nursing carry out nursing care at various levels of healthcare. Work-life quality refers to an individual's feelings concerning work and outcomes and depends on different working characteristics and conditions. Quantitative research based on a cross-sectional study was used. This cross-sectional study included 486 employees in nursing from four Slovenian acute care hospitals. The results showed that most employees in nursing assessed the work-life quality on a moderate level: 76% were satisfied with their work, and 89% assessed their well-being at the workplace as positive. Considering the leaders' support, the number of patients, adequate information, teamwork, working position, use of days off, and equipment for safe work, we can explain the 53.5% of the total variability of work-life quality. We also found that work-life quality had an essential effect on well-being at the workplace (β = 0.330, $p < 0.001$) and work satisfaction (β = 0.490, $p < 0.001$) of employees in nursing. Work-life quality refers to an employees' feelings about their workplace, and its monitoring is important for higher employees' well-being and health. For management and policymakers in nursing, it is important to design strategies to ensure an adequate number of competent employees and establish a supportive leadership system. Work-life quality is an important factor in the recruitment and retention of the nursing workforce. Flexible working conditions and policy changes can improve work-life quality and balance. Nursing management must understand the influencing factors of work-life quality to improve nursing employee retention strategies.

**Keywords:** hospital; work-life quality; nursing; well-being; COVID-19

## 1. Introduction

COVID-19 affected the work-life ratio of almost every person in the world. Remote work has become a central way of working for many employees [1], while the situation has been completely different for health professionals. The quality of work and life at the workplace of healthcare employees was influenced by "human connection" and "mastery" despite exceptional workloads [2]. Improving the work-life quality of nurses, reducing turnover and the number of people leaving the profession, and, as a result, ensuring organizational stability are some of the most challenging issues today [3]. Work-life quality refers to the quality that an individual feels concerning their work and outcomes [4]. In healthcare, we have faced an ever-decreasing number of employees in nursing and an increase in the workload of employees [4,5], which affects the work-life quality, specifically in nurses working with patients with COVID-19 [6,7]. Almost one out of three employees in nursing thought of quitting their job during the restrictions. It is known that employees in nursing's intention to leave their profession is related to work-life quality [7–9]. Most employees in nursing thought their work-life quality changed negatively during the COVID-19 restrictions [7]. During the COVID-19 restrictions, employees in nursing faced with problems such as compassion of fatigue [10], depression [11], fear of being unable

to help patients, uncertainties about the treatment and course of the disease and limited clinical knowledge [12], stigmatization due to the risk of infection [13], and a changing work routine and lifestyle, which have negatively affected healthcare professionals' quality of life. Before the COVID-19 restrictions, nursing was already known to be a demanding job, and work stresses can have different adverse effects [9].

Research shows significant factors affecting work-life quality are hospital level, age, income, night shift attendance, the patient-to-nurse ratio [13], and balanced work–family needs [14]. They are also associated with the work-life quality of rotating shifts [15], working time, lack of staff and materials, the ability to take the initiative and workplace safety [16], the demands of a hard job, and work stress [17,18]. Work-life quality for employees in nursing is often overlooked as most work involves dealing with patients' quality of life.

Considering the new challenges nursing employees faced during the COVID-19 restrictions and the growing global problem of nursing staff shortages, it is important to research how and to what extent workplace factors can contribute to higher well-being at the workplace and higher work-life quality. The aim was to examine the relationship between work-life quality and workplace factors of employees in nursing working in hospitals during the COVID-19 restrictions.

## 2. Literature Review

### 2.1. Work-Life Quality

The relationship between employees and the overall working environment is defined as the quality of working life. The World Health Organization defined quality of life as an individual's perception of their position in life in the context of the culture and value systems in which they live and in relation to their goals, expectations, standards and concerns [19]. Work-life quality has become an important concept; as the largest profession employed in hospitals, nursing employees are vital to healthcare, and employee productivity is still a frequent topic of research [20]. A decade ago, the work-life quality of nursing employees was investigated to assess whether employees fulfill their individual needs through experience and simultaneously fulfill the organization's goals [21]. Research shows that work-life quality is related to employees' emotional, physical, and well-being and, consequently, to the organization's results, the quality of work, and turnover. Characteristics of work, mental and physical well-being, and health characteristics, organizational characteristics, balance between private and work environment, and professional identity are key components of work-life quality [21,22].

Factors cited by nursing employees as having a significant impact on the work-life quality include infrastructure [23], lack of time for breaks [24], lack of incentives in work organization [25], poor social and collegial support [26], role conflict in the team [27], night shifts, overtime work, work planning and scope of work obligations, recognition of the work performed, support, providing autonomy in the work, sufficient staffing and working conditions [28].

Developing an environment where employees feel supported and valued allows them to find the power to balance their personal and professional lives and is of the utmost importance. Improving the work-life quality of employees in nursing can also be achieved by ensuring organizational stability and reducing turnover [29].

The consequences of low work-life quality are anger shock, shame-somatization, depression, anxiety, nightmares, loss of appetite and changing jobs or professions [30], aggravated medicine consumption, sleep disorders, and headaches [31]. The research shows a lower work-life quality to be related to the social sphere, affecting the quality of medical care [32]. Work satisfaction is often associated with compassion, burnout, and traumatic stress, thus affecting work-life quality. Work-life quality is also influenced by stress and job satisfaction [33].

The restrictions significantly impacted already strained healthcare systems, and clinical environments still face extraordinary challenges when trying to provide health care

and ensure the quality of professional and private life for sustainable management in nursing [34]. Lack of work-life quality for employees in nursing has resulted in staff shortages, forcing many hospitals to close wards or reduce the number of beds [35].

### 2.2. Well-Being at the Workplace

The workplace is the place we create within which people come together to perform their work and achieve results. The workplace can be defined as a psychological climate affecting the individual's well-being [36]. The connection between the employees and the environment determines the psychological and social dimensions of this environment, which shows how the individual feels in this workplace [37].

The state of physical and mental health, which is reflected in the joy and sense of the profession, work, professional satisfaction, and engagement at work, is the result of well-being at the workplace [38]. Lower well-being at the workplace in nursing affects the employees, patients, healthcare organizations, and the wider society [39]. Organizational factors of well-being at the workplace are teamwork [40], pride in work, sense of mission, social integration [41], and stressful situations [42]; in addition, leaders play a key role in the work satisfaction of employees in nursing, which, in turn, affects patient satisfaction [43].

Well-being at the workplace of employees in nursing during the COVID-19 restrictions was negatively affected by the work–life balance [38].

Research establishes a connection between work-life quality and well-being at the workplace [37]. At the workplace, employees in nursing face a lack of teamwork, a culture of blame and fear, lack of management support, poor communication, bullying, equipment problems, and an inability to share expertise or make decisions [38]. Poor well-being at the nursing workplace has a negative impact on the quality of nursing care and the effectiveness of nursing employees [39], professional relationships, and overall work performance, especially during a pandemic [43]. The workplace in nursing is also significantly related to increased work efficiency, patient safety, and quality of care [44] because managers in nursing cannot change work, but they can influence the creation of a positive workplace that supports job satisfaction and well-being at the workplace [44,45]. After reviewing the literature, we can report that during the COVID-19 pandemic, there were almost no studies examining the work-life quality of employees in nursing. The situation during the COVID-19 pandemic was uncertain and had the greatest impact on the healthcare systems; therefore, the healthcare workers faced daily changes, fear, and uncertainty. To identify research gaps and guide the analysis, the research drew on two literature streams that inform management about the quality of professional and private life under the COVID-19 restrictions for sustainable development. Given the above, we decided to investigate to what extent measures and changes in the functioning of the healthcare system affect the quality of working life to guide management. The present study was broad in scope; it was not only focused on biological characteristics (age, gender, education, working experiences, and marital status) but also on characteristics related to restrictions of the COVID-19 pandemic (number of patients, accessibility to information, adequate equipment for work, teamwork, use of days off, exposure to stress, managing stress, collaboration), and workplace characteristics (leaders' support, well-being at the workplace), because we wanted to ascertain the possible number of factors that influence the work-life quality of employees in nursing. Because the leaders' support is essential according to characteristics in nursing related to COVID-19 restrictions, we hypothesized:

**H1.** Leaders' support is related to work-life quality.

**H2.** Work-life quality affects the well-being at the workplace.

### 3. Materials and Methods

#### 3.1. Setting and Participants

After more than 18 months of COVID-19, when the restrictions significantly interfered with and changed the organization of work in a healthcare institution, we decided to

investigate the work–life balance of employees in nursing in Slovenian hospitals from October to December 2021.

Four invited hospitals that cared for COVID-19 patients from different regions of Slovenia participated in the research. Two hospitals were tertiary-level hospitals, and two were secondary-level hospitals. In each participating hospital, 300 questionnaires were distributed. The research coordinators in the participating hospitals distributed the questionnaires during the morning shift. The questionnaires were collected in special design cases in a pre-determined area to ensure anonymity. Questionnaires ($n$ = 2000) were delivered to the participating hospitals, 484 of which were completed. The response rate was 40%. The mean age was 39.5 ± 10.6 years (95%, CI = 38–44), and the mean working experience was 16.8 ± 11.6 years (95%, CI = 15.2–18.5). In hospitals on a secondary level, the mean age was 40.14 ± 10.4 years (95%, CI = 38.2–42.2), and the mean working experience was 17.63 ± 11.6 years (95%, CI = 15.5–19.9). In hospitals on a tertiary level, the mean age was 38.78 ± 10.8 years (95%, CI = 37.7–39.9), and working experience time was 15.92 ± 11.7 years (95%, CI = 14.8–17.1). A total of 83 (17%) respondents were men, and 401 (83%) were female. In hospitals on a secondary level, 15.5% (17) were males and 84.5% (93) females, and in hospitals on a tertiary level, 17.6% (66) were males and 82.4% (308) females. In hospitals on a secondary level, 48.2% (53) employees finished secondary health school, 45.4% (50) completed a bachelor's degree, and 6.4% (7) finished postgraduate studies. In hospitals on a tertiary level, 34.5% (129) employees finished secondary health school, 52.6% (197) completed a bachelor's degree, and 12.8% (48) finished postgraduate studies.

*3.2. Data Collection*

A questionnaire with closed-type questions was used in the research. The first part of the questionnaire included demographic questions (gender, level of education, working years, etc.) followed by items from workplace factors (work–life balance, exposure to stress, managing stress, teamwork, etc.), well-being at the workplace, work satisfaction and work-life quality. Well-being at the workplace was assessed using the wellbeing scale [21], which contains 54 items. A sum of the higher scores indicates a higher level of workplace well-being. The sum of the scale ranged from 54 to 270. Cronbach's alpha was 0.991. Work-life quality was assessed using the quality of work life questionnaire, which contains 24 items, following the Likert-type five-point answer format of: 1 (Never); 2 (Rarely); 3 (Moderately); 4 (Frequently); and 5 (Always). Twenty-four items were related to the eight dimensions of work-life quality: safety and health in working conditions (six items), fair and adequate compensation (two items), opportunity for use and capacity development (two items), constitutionalism in the organization of work (two items), social integration at work (two items), career opportunities and security (two items), social relevance of life at work (three items), and work and total living space (five items) [46]. Cronbach's alpha was 0.789.

*3.3. Statistical Analysis*

A descriptive analysis was performed to assess the work-life quality and workplace factors in nursing. The Kolmogorov–Smirnov test confirmed that the studied variables were not normally distributed ($p$ < 0.001). The differences between groups were compared with the Mann–Whitney U-test and the Kruskal–Wallis H-test. The Spearman correlation coefficient was used to establish possible correlations and the regression analysis to determine the impact of studied independent variables on work-life quality (dependent variables). A $p$-value of <0.05 was considered statistically significant. Statistical analysis was performed with SPSS version 27.0 (SPSS Inc., Chicago, IL, USA).

**4. Results**

Regarding the workplace, in hospitals on the secondary level, 57.3% ($n$ = 63, 95%, CI = 47.3–66.4) of employees in nursing are satisfied, and 9.1% ($n$ = 10, 95%, CI = 3.6–14.5) are very satisfied with their job. A total of 60% ($n$ = 66, 95%, CI = 50–69.1) are satisfied, and

20% (*n*= 22, 95%, CI = 13.6–28.2) are very satisfied with leaders' support. A total of 33.6% (*n* = 37; 95%, CI = 24.5–42.7) are very often and 43.6% (*n* = 48, 95%, CI = 34.5–52.7) are often exposed to stress. A total of 56.4% (*n* = 211; 95%, CI = 51.1–61.2) employees in nursing in hospitals on the tertiary level are satisfied, and 15% (*n* = 56; 95%, CI = 11.5–18.4) are very satisfied with their job; in addition, 50% (*n* = 187; 95%, CI = 44.7–55.1) are satisfied, 50% (*n* = 187; 95%, CI = 44.9–55.1) are satisfied, and 29.7% (*n* = 111, 95%, CI = 25.4–34.8) are very satisfied with leaders' support. In tertiary hospitals, 28.1% (*n* = 374; 95%, CI = 23.8–32.6) are very often exposed to stress and 42.5% (*n* = 159; 95%, CI = 37.4–47.6) are often exposed to stress.

According to the demographic data, there were no significant differences between groups according to gender ($Z = 0.536$, $p = 0.592$), age ($Z = 1.160$, $p = 0.246$) and working experiences ($Z = 1.290$, $p = 0.197$). We found significant differences only in the level of education ($Z = 2.835$, $p < 0.001$) between nursing employees in hospitals on the secondary and tertiary levels.

We found significant differences in work–life balance ($Z = 4.291$, $p < 0.001$), use of days off ($3.785$, $p < 0.001$), sick leave ($Z = -2.731$, $p = 0.003$), number of patients in all shifts, and also during the weekend ($Z = 2.686$, $p = 0.007$), well-being at the workplace ($Z = 5.418$, $p < 0.001$) and work-life quality ($Z = 2.224$, $p = 0.045$) between nursing employees in hospitals on the tertiary and secondary level (Table 1).

**Table 1.** Results for work-life quality and workplace factors.

| Variables | Tertiary Hospitals | | Secondary Hospitals | | Z | p |
|---|---|---|---|---|---|---|
| | $\bar{x} \pm s$ | 95%CI | $\bar{x} \pm s$ | 95% CI | | |
| Work–life balance | 3.02 ± 0.88 | 2.8–3.1 | 2.51 ± 1.00 | 2.4–2.6 | −4.291 | <0.001 |
| Leaders' support | 4.07 ± 0.79 | 3.9–4.2 | 3.95 ± 0.76 | 3.8–4.1 | −1.1328 | 0.184 |
| Use days off | 3.97 ± 0.87 | 3.8–4.2 | 3.57 ± 0.64 | 3.3–3.8 | −3.785 | <0.001 |
| Equipment for safety work | 3.15 ± 0.37 | 2.9–3.5 | 3.14 ± 0.36 | 2.9–3.4 | −0.245 | 0.800 |
| Sick leave | 3.60 ± 0.76 | 3.4–0.7 | 2.90 ± 0.9 | 2.5–3.3 | −2.731 | 0.003 |
| Work satisfaction | 3.78 ± 0.86 | 3.7–3.9 | 3.64 ± 0.82 | 3.4–4.0 | −1.629 | 0.103 |
| Exposure to stress | 3.90 ± 0.89 | 3.8–4.0 | 4.07 ± 0.83 | 3.6–4.2 | 1.568 | 0.117 |
| Managing stress | 3.67 ± 0.73 | 3.6–3.8 | 3.56 ± 0.78 | 3.2–3.9 | −1.304 | 0.192 |
| Teamwork | 3.87 ± 0.87 | 3.8–4.0 | 3.75 ± 0.80 | 3.6–3.9 | −1.238 | 0.216 |
| Effective communication at work | 2.96 ± 0.97 | 2.8–3.1 | 2.82 ± 0.84 | 2.2–2.9 | −1.090 | 0.276 |
| Number of patients—morning | 12.72 ± 9.46 | 11.7–13.7 | 17.03 ± 11.52 | 14.7–19.5 | 3.840 | <0.001 |
| Number of patients—afternoon | 16.85 ± 11.53 | 15.6–18.1 | 23.07 ± 16.74 | 19.7–36.7 | 3.288 | 0.001 |
| Number of patients—night | 16.78 ± 11.87 | 15.5–17.9 | 21.73 ± 16.55 | 12.6–15.8 | 2.821 | 0.005 |
| Number of patients—weekend | 16.73 ± 11.67 | 15.4–17.9 | 24.47 ± 2187 | 20.4–29.4 | 2.686 | 0.007 |
| Well-being at workplace | 4.02 ± 0.59 | 3.9–4.2 | 3.66 ± 0.62 | 3.6–3.7 | −5.418 | <0.001 |
| QWL—safety and health | 3.19 ± 0.43 | 3.14–3.32 | 3.11 ± 0.54 | 3.1–3.2 | −0.084 | 0.333 |
| QWL—work and living space | 3.14 ± 1.21 | 2.9–3.4 | 2.93 ± 0.65 | 2.9–3.0 | −2.336 | 0.019 |
| QWL—organisation of work | 3.25 ± 0.73 | 3.1–3.4 | 2.97 ± 0.80 | 2.9–3.1 | −3.238 | 0.001 |
| QWL—fair compensation | 3.13 ± 1.06 | 2.9–3.3 | 2.92 ± 0.93 | 2.8–3.0 | −2.277 | 0.023 |
| QWL—career opportunities | 2.95 ± 0.89 | 2.8–3.1 | 2.89 ± 0.86 | 2.8–3.0 | −0.694 | 0.488 |
| QWL—develop human capabilities | 2.70 ± 0.80 | 2.5–2.9 | 2.75 ± 0.92 | 2.6–2.9 | 0.090 | 0.488 |
| QWL—social relevance | 3.57 ± 0.74 | 3.4–3.7 | 3.53 ± 0.93 | 3.4–3.6 | −2.162 | 0.038 |
| QWL—social integration | 2.97 ± 0.85 | 2.8–3.1 | 2.94 ± 0.80 | 2.9–3.0 | −0.473 | 0.336 |
| Work-life quality—QWL | 3.22 ± 0.48 | 3.0–3.3 | 2.99 ± 0.48 | 2.9–3.1 | — | — |
| QWL—Total | 77.2 ± 9.75 | 74.9–80.8 | 71.76 ± 9.19 | 70.1–76.5 | −2.224 | 0.045 |
| QWL—Min-max | 45.0–111.2 | | 51.2–97.2 | | — | — |

From Table 2, we can see that all ten studied workplace factors are related to work-life quality. Social relevance is related to all ten studied workplace factors, and a fair work-life quality and adequate compensation are associated with nine of ten studied workplace

factors. Adequate equipment for safety work, leaders' support, and work satisfaction are related to seven of the eight dimensions of work-life quality.

**Table 2.** Results of Spearman correlation test for dimensions of work-life quality and other workplace factors.

| Variables | QWL-S | QWL-W | QWL-C | QWL-F | QWL-CA | QWL-O | QWL-S | QWL-SI | QWL-TOTAL |
|---|---|---|---|---|---|---|---|---|---|
| WP | 0.066 | 0.041 | 0.115 ** | 0.170 ** | 0.056 | 0.026 | 0.108 * | 0.093 * | 0.102 * |
| EQ | 0.160 ** | 0.105 * | 0.104 * | 0.241 ** | 0.209 ** | 0.118 ** | 0.039 | 0.134 ** | 0.103 * |
| INF | 0.059 | 0.097 * | 0.075 | 0.137 ** | 0.088 * | 0.015 | 0.112 * | 0.099 * | 0.153 ** |
| WBW | 0.252 ** | 0.081 | 0.130 ** | 0.232 ** | 0.278 ** | 0.152 ** | 0.047 | 0.277 ** | 0.165 ** |
| WLB | 0.103 * | 0.030 | 0.041 | 0.208 ** | 0.070 | 0.016 | 0.185 ** | 0.064 | 0.116 ** |
| LS | 0.133 ** | 0.060 | 0.253 ** | 0.358 ** | 0.346 ** | 0.135 ** | 0.107 * | 0.223 ** | 0.251 ** |
| WS | 0.226 ** | 0.029 | 0.164 ** | 0.312 ** | 0.332 ** | 0.096 * | 0.219 ** | 0.244 ** | 0.244 ** |
| ES | −0.224 ** | −0.104 * | −0.043 | −0.138 ** | −0.062 | −0.123 ** | −0.049 | −0.164 ** | −0.107 * |
| MS | 0.081 | 0.052 | 0.101 * | 0.116 ** | 0.191 ** | 0.091 * | 0.128 ** | 0.124 ** | 0.164 ** |
| TW | 0.195 ** | 0.069 | 0.0154 ** | 0.245 ** | 0.238 ** | 0.020 | 0.093 * | 0.145 ** | 0.139 ** |

Legend: ** = Correlation is significant at the 0.01 level (two-tailed); * = Correlation is significant at the 0.05 level (two-tailed); QWL-S = work-life quality—safety and healthy working conditions; QWL-W = work-life quality—work and total living space; QWL-C = work-life quality—constitutionalism; QWL-F = work-life quality—fair and adequate compensation; QWL-CA = work-life quality—career opportunities and security; QWL-O = work-life quality—opportunity for use; QWL-S = work-life quality—safety and healthy working conditions; QWL-SI = work-life quality—social relevance at work; WP = working position; EQ = equipment for safety work; INF = adequate information for work; WBW = well-being at the workplace; SL = sick leave; WLB = work–life balance; LS = leaders' support; WS = work satisfaction; ES = exposure to stress; MS = managing stress; TW = teamwork; QWL = work-life quality.

We were interested in which of the studied workplace factors affect the work-life quality in nursing. With the regression analysis (Table 3), we can explain 53.5% of the total variability of the work-life quality of employees in nursing through the type of institutions, working positions, leaders' support, equipment for safety work, adequate information, teamwork, number of patients and use of days off.

**Table 3.** Results of the regression analysis for the work-life quality.

| | Variables | *B* | SE | β | *t* | *p* |
|---|---|---|---|---|---|---|
| | Institution | 0.090 | 0.027 | 0.106 | 2.228 | 0.026 |
| | Working positions | 0.072 | 0.033 | 0.096 | 2.207 | 0.028 |
| | Equipment for safety work | 0.121 | 0.032 | 0.168 | 3.811 | <0.001 |
| | Leaders' support | 0.073 | 0.032 | 0.116 | 2.353 | 0.019 |
| $R^2 = 0.535$ | Exposure to stress | −0.027 | 0.025 | −0.049 | −1.264 | 0.207 |
| | Adequate information's | 0.190 | 0.061 | 0.136 | 3.107 | 0.002 |
| | Teamwork | 0.120 | 0.037 | 0.108 | 2.147 | 0.018 |
| | Sick leave | −0.032 | 0.030 | −0.049 | −1.069 | 0.286 |
| | Number of patients | −0.003 | 0.002 | −0.092 | −2.649 | 0.008 |
| | Use days off | 0.044 | 0.023 | 0.101 | 2.044 | 0.042 |

Legend: $R^2$—coefficient of determination; *B*—unstandardized regression coefficient; SE—standard error; β—standardized regression coefficient; *t*—a value of t-statistic; *p*—a significance value.

Using linear regression analysis, we also found that work-life quality affects well-being at the workplace (β = 0.330, t = 5.839, *p* < 0.001) and work satisfaction (β = 0.490, t = 6.410, *p* < 0.001) of employees in nursing. With work-life quality, we can explain 25% of the total variability of well-being at the workplace and 27% of the total variability of work satisfaction of nursing employees.

We did not find significant differences in work-life quality ($Z = -0.022$, $p = 0.823$) between men and women. We found significant differences in well-being at the workplace ($Z = 2.000$, $p = 0.046$), exposure to stress ($Z = 3.139$, $p = 0.002$), and work–life balance ($Z = 2.704$, $p = 0.033$) according to gender. According to the level of education (($\chi 2(2) = 8.835$, $p = 0.045$) and working position (($\chi 2(2) = 8.572$, $p = 0.036$), we found significant differences in work-life quality. There were no significant differences in work-life quality according to marital status (($\chi 2(2) = 1.962$, $p = 0.375$), the number of children (($\chi 2(2) = 1.163$, $p = 0.452$), and the number of days of sick leave (($\chi 2(2) = 4.290$, $p = 0.232$). Differences in work-life quality were found between those who assessed that they had or did not have safe work equipment ($Z = 5.033$, $p < 0.001$), between those who assessed that leaders understand or do not ($Z = 4.032$, $p < 0.001$), and those who obtained all information for work or not ($Z = 3.247$, $p = 0.001$). There is no difference in work-life quality between those who worked full-time or part-time ($Z = 0.320$, $p = 0.749$) and between those who worked different shifts (($\chi 2(2) = 4.482$, $p = 0.106$).

## 5. Discussion

The results showed that most employees in nursing assessed the work-life quality on a moderate level during the COVID-19 restrictions. In addition, some other studies have shown that employees in nursing assessed work-life quality at a moderate level during the COVID-19 restrictions [47] and before the COVID-19 restrictions [48]. The results of the work-life quality of the employees in nursing in Slovenian hospitals are encouraging in comparison with similar studies during the COVID-19 restrictions; the level of work-life quality is on the same level of Iranian employees in nursing [49] and higher by 3% in comparison with the study of Korean employees in nursing from tertiary general hospitals [18]. Our positive results revealed that only one-quarter (24%) of employees in nursing were dissatisfied with their job and 11% were dissatisfied with the work-life quality. Because our research was conducted during the COVID-19 restrictions, we can say that our results are very supportive in comparison with the results of the previous studies before the COVID-19 restrictions in Taiwan [9], Saudi Arabia [50], Sweden [51], and in seven European countries [52]. Baysal et al. [7] noted that most employees in nursing thought that their work-life quality negatively changed during the COVID-19 restrictions. This study provides an initial step in understanding the work-life quality of employees in nursing hospitals at the secondary and tertiary levels during extraordinary circumstances, such as COVID-19 restrictions, making it even more important.

We did not find significant differences in work-life quality according to marital status and the number of children. Our results contradict those of Tanaka et al. [53], who found that work gap scores and family gap scores for employees in nursing living alone were significantly higher, respectively, than those working in nursing and living with family. They also found that the work–life balance gap was associated with employees' quality of life. In our research, we also found that work–life balance is related to work-life quality. On the other hand, Lebni et al. [47] found that the work-life quality in public hospitals significantly correlated with respondent age, marital status, education, work experience, position, department, shifts, and employment status. Our research, which was also carried out in public hospitals, found that work-life quality was not associated with age, marital status, level of education and working experiences. We found that work-life quality is associated with the availability of equipment for safe work, adequate information, leaders' support, teamwork, and exposure to stress. According to these findings, our results are supportive because we found that employees in nursing were very satisfied with leaders' support; 90% received adequate information, 99% assessed that they had enough equipment for safety work, 97% felt safe about their work, and 96% had a person who was available for all the information.

According to the results, leaders' support, equipment for safety work and adequate information are the most important factors affecting work-life quality. With the regression analysis, we found that with previous factors together with number of patients, teamwork,

working position and use the days off, we can explain the 53.5% of the total variability of work-life quality. In their qualitative systematic review, Joo et al. [54] identified five barriers to COVID-19 care in hospital-based employees in nursing. These barriers are comparable with our results, and they are: limited information about COVID-19, unpredictable tasks and challenging practice, insufficient support, concern about family, and stress. Our results complement the results of Wang et al. [13], who found that stress, hospital level, age, income, night shift, and patient-to-nurse ratio are significant factors affecting work-life quality. This is also in comparison with Inocian et al. [33], who noted that work-life quality is influenced by stress and work satisfaction.

We also found that work-life quality had an essential effect on well-being at the workplace and the work satisfaction of employees in nursing. Good assessment of work-life quality is also related to other encouraging results, such as 76% of employees in nursing being satisfied with their work, and 89% having assessed their well-being at the workplace as positive. These encouraging results can be linked to an excellent evaluation of the leaders' support and leaders' understanding by the nursing employees in hospitals at secondary and tertiary levels. In hospitals on both levels, we did not find differences in teamwork, leaders' support, effective communication, equipment for safe work, exposure to stress, and managing stress. We agree with Gab et al. [55], who noted that nursing leaders have an extremely important task of showing support for nursing work, as this is an important factor in nurses' job satisfaction; they must simultaneously build teamwork and provide support to colleagues. It is important that the organization knows that to increase work-life quality, it should ensure the well-being at the workplace, and not only focus on time related to work but on balancing personal life and work. According to the results, we can say that nursing management recognized the importance of supporting and encouraging employees in nursing to achieve better outcomes for employees and patients in hospitals at different levels, which is of the utmost importance, especially in times of extraordinary circumstances (which the COVID-19 restrictions certainly are).

Between employees in nursing from hospitals at different levels, there were no significant differences in work satisfaction; however, differences were found in well-being at the workplace, sick leave, use of days off, and work–life balance between secondary and tertiary level hospitals. Most of these differences result the organization of work in hospitals on different levels according to the different number of employees, level of education, possibility of using days off, etc. In addition, Jonker et al. [56] noted differences between hospitals according to the quality of patient information, a higher degree of observed staff teamwork, more confidence, and a better overall inpatient experience, which were also significantly correlated. We can also connect these differences between hospitals' levels with an ever-decreasing number of nursing employees, an increasing number of patients, and an increase in the workload of employees, especially during the COVID-19 restrictions (which was also found in previous research [4,5]). We agree with Niu et al. [6], who noted that the shortage of employees in nursing affects the work-life quality, specifically for employees who are fighting COVID-19.

Healthcare management must be aware that constantly monitoring and improving employees' work-life quality in nursing is necessary because work-life quality is associated with well-being at the workplace and work satisfaction. These results are in comparison with the results from Slovenia, which reported that work-life quality is associated with the psychological well-being and mental health of employees in nursing [57]. We also agree with Abdesalam et al. [58], who noted that healthcare system reforms are required to improve work-life quality and well-being at the workplace in nursing.

Helping to achieve good work-life quality in hospitals at all levels can promote nursing employee recognition, which might emerge as a valuable resource and health policy tool. A positive climate, with appropriate leaders' support and adequate monitoring of workplace factors, may lead to a better work-life quality with less distress and overwhelming circumstances for employees in nursing. Nursing managers can reduce nursing employees' intention to leave the job by providing safe, comfortable, accessible, and appropriate

working conditions, positive change supporting employees in nursing, mastering interpersonal engagement skills, and creating trust. Understanding the influencing factors of work-life quality is also important for nursing management to improve nursing employee retention strategies.

There are also some limitations in the study. First, this study only investigated the employees in nursing from hospitals at the secondary and tertiary levels. Second, this cross-sectional study only provided information at a single point. Nevertheless, it is of the utmost importance to take care of healthcare workers due to the time (the COVID-19 restrictions) in which the research was conducted. The overall literature review of work-life quality research showed that work-life quality before the COVID-19 pandemic included some factors considered personal feelings or perceptions toward work, organization, and employers. In the future, we recommend a qualitative study to understand work-life quality in nursing. The results of this studies can be used as a basis for developing context-based instruments for measuring the work-life quality in nursing. Nerveless, this work has a major contribution to current research and practice for sustainable management and healthcare not only in the Slovenian area and not only during the COVID-19 restriction period; therefore, this study is a data driver.

## 6. Conclusions

The present study complements the current literature on nursing employees' work-life quality and understanding of the work-life quality among employees working in hospitals on different levels. The study analyses the components of management based in the organizations to provide better work-life quality for employees in nursing.

The work-life quality rate and factors are influenced by variables reflecting the organization of work, skills, and nursing management competencies for work with employees in nursing. The results of the study support the need for nursing leadership and for policymakers to prepare activities to ensure a sufficient number of competent employees in nursing, competitive organizations and to encourage a healthy work environment. Only an appropriate management approach with positive attitudes and behavior is effective in increasing the nursing employees' workability and healthy lifestyle behaviors affecting their work-life quality and quality of life. Improving working standards, implementing strategies that enhance work environment, better organization work, and a sufficient number of competent nursing employees is necessary to achieve quality and higher patient and employees' work satisfaction and well-being at the workplace. Healthcare management should create and maintain a work environment that fosters and supports nursing employees' decision-making ability and is more flexible, which might prevent employees in nursing from working extra hours, being overstressed, or leaving the profession.

Ultimately, we can say that work-life quality is an important factor in the recruitment and retention of the nursing workforce. The improvement of work-related quality of life helps reduce absenteeism and improves productivity. It is necessary to develop an efficient strategy, such as establishing a support system for nursing employees and leaders, and improving teamwork.

**Author Contributions:** Conceptualization, M.D. and M.L.; methodology, M.L.; software, M.L.; validation, M.L.; formal analysis, M.L.; investigation, M.D.; resources, M.L. and M.D.; data curation, M.L. and M.D.; writing—original draft preparation, M.L. and M.D.; writing—review and editing, M.L.; visualization, M.D.; supervision, M.L.; project administration, M.L. and M.D; funding acquisition M.D. All authors have read and agreed to the published version of the manuscript.

**Funding:** This research received no external funding.

**Institutional Review Board Statement:** The research includes human data which are in accordance with the Declaration of Helsinki and has been approved by the Ethics Committee (No. 04/1R-2020).

**Informed Consent Statement:** Informed consent was obtained from all subjects involved in the study.

**Data Availability Statement:** Not applicable.

**Conflicts of Interest:** The authors declare no conflict of interest.

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
