# Peer review of "The Importance of Monitoring the Work-Life Quality during the COVID-19 Restrictions for Sustainable Management in Nursing"

_sustainability, doi:10.3390/su15086516_

Round 1

Reviewer 1 Report

The manuscript provides a discussion about the importance of work-life balance in a Covid-19 scenario among nursing staff. The following points are observed. 

1. The manuscript is well organized and easy to follow. 

2. A questionnaire based cross sectional study including 486 participants was conducted and the results signifies the importance of work-life balance among nursing staff working at different levels of hospitals during the Covid-19 pandemic. 

3. The study identifies the positive role of leadership for promoting positive work-life balance and how the leadership can impact this balance to promote a culture of positive well-being among the nursing staffs working at different levels. 

4. It is suggested to include few more latest state-of-the art on the topic and provide a comparison on how the proposed study adds strength to the already existing literature. 

Author Response

We thank the editor and all reviewers for their feedback on our manuscript. According to the reviewers' comments, we have prepared the revised manuscript “The importance of monitoring the work-life quality during the COVID-19 restrictions for sustainable management in nursing”. We have revised our manuscript following each raised point and carefully considered all the reviewers' comments. Major corrections have been performed and made English proofreading (added a certificate).

After the review of the English proofreader, we changed the word COVID-19 pandemic to COVID-19 restrictions throughout the entire manuscript - this also led to a minor change in the title.

In the paragraphs below, we list reviewers’ comments, followed by our responses addressed to a specific comment.

All changes in the manuscript we made by Track changes.  

Best regards,

Mateja Lorber, Mojca Dobnik

Reviewer 1:

Comments 1, 2, 3:

The manuscript provides a discussion about the importance of work-life balance in a Covid-19 scenario among nursing staff. The following points are observed. 

  1. The manuscript is well organized and easy to follow. 
  2. A questionnaire based cross sectional study including 486 participants was conducted and the results signifies the importance of work-life balance among nursing staff working at different levels of hospitals during the Covid-19 pandemic. 
  3. The study identifies the positive role of leadership for promoting positive work-life balance and how the leadership can impact this balance to promote a culture of positive well-being among the nursing staffs working at different levels. 

Responses 1,2,3: We thank the reviewer for all the positive comments.

Comment 4: It is suggested to include few more latest state-of-the art on the topic and provide a comparison on how the proposed study adds strength to the already existing literature.

Response 4: We thank the reviewer for the comment, and, as suggested, we added some of the latest results to the discussion and present how our results strengthen existing literature.

Reviewer 2 Report

The topic is interesting. The aim of the study should be clearly expressed. Thereby, theoretical bakcground and literature review should be deeple investigated and documented.

Author Response

We thank the editor and all reviewers for their feedback on our manuscript. According to the reviewers' comments, we have prepared the revised manuscript “The importance of monitoring the work-life quality during the COVID-19 restrictions for sustainable management in nursing”. We have revised our manuscript following each raised point and carefully considered all the reviewers' comments. Major corrections have been performed and made English proofreading (added a certificate).

After the review of the English proofreader, we changed the word COVID-19 pandemic to COVID-19 restrictions throughout the entire manuscript - this also led to a minor change in the title.

In the paragraphs below, we list reviewers’ comments, followed by our responses addressed to a specific comment.

All changes in the manuscript we made by Track changes.  

Best regards,

Mateja Lorber, Mojca Dobnik

Reviewer 2:

Comment 1: The topic is interesting. The aim of the study should be clearly expressed. Thereby, theoretical background and literature review should be deeple investigated and documented.

Responses 1: We thank the reviewer for the comment and, as suggested, have presented an aim and restructured the theoretical background.

Reviewer 3 Report

This article discusses nurses' work-life quality and related workplace factors during COVOD-19. First, there is no theoretical support for the relationship between variables, so it is only verified by evidence from statistical analysis of data. Furthermore, during the period of COVOD-19, many studies have explored the relevant work variables of nurses, resulting in the low incremental contribution of this paper.

Introduction

1. First of all, every expression in the introductory paragraph should be supported by citing literature unless the theory of the paragraph is created by the author himself. The content of the first paragraph of the introduction is some general knowledge or market research data, and it is impossible to see what academic gap and importance this research fills. The same problem is the literature paragraph.

2. Second, this is no “literature review”, this format is not compatible with social science research.

Materials and Methods

1.      Need to justify why regression analysis was used rather than other methods of analysis such as structural equation modeling

Discussion

1.      You only intruded on the analysis results but you should cite past studies to verify the consistency. 

2.      I suggest you should rewrite the discussion to demonstrate more contributions.

Author Response

We thank the editor and all reviewers for their feedback on our manuscript. According to the reviewers' comments, we have prepared the revised manuscript “The importance of monitoring the work-life quality during the COVID-19 restrictions for sustainable management in nursing”. We have revised our manuscript following each raised point and carefully considered all the reviewers' comments. Major corrections have been performed and made English proofreading (added a certificate).

After the review of the English proofreader, we changed the word COVID-19 pandemic to COVID-19 restrictions throughout the entire manuscript - this also led to a minor change in the title.

In the paragraphs below, we list reviewers’ comments, followed by our responses addressed to a specific comment.

All changes in the manuscript we made by Track changes.  

Best regards,

Mateja Lorber, Mojca Dobnik

Reviewer 3:

Comments 1, 2 - Introduction: First of all, every expression in the introductory paragraph should be supported by citing literature unless the theory of the paragraph is created by the author himself. The content of the first paragraph of the introduction is some general knowledge or market research data, and it is impossible to see what academic gap and importance this research fills. The same problem is the literature paragraph. 2. Second, this is no “literature review”, this format is not compatible with social science research.

Responses 1,2: We thank the reviewer for the comments and as suggested, we added a theoretical background, presented a gap, and all the expressions in the introductory paragraph are supported by the literature.

Comment 3: Need to justify why regression analysis was used rather than other methods of analysis such as structural equation modeling

Response 3:. We thank the reviewer for the comment. We used a regression analysis as a mere prediction with the reason for including several predictors mostly informational. Our question was does a predictor explain variance beyond the inclusion of others. It was a preliminary study that will continue and include additional variables and more advanced statistical analysis.

Comments 4,5:. 1.You only intruded on the analysis results but you should cite past studies to verify the consistency. 2 I suggest you should rewrite the discussion to demonstrate more contributions.

Responses 4,5: We thank the reviewer for the comment; as suggested, we added other studies, compared them with our results, and rewrote a discussion.

Reviewer 4 Report

It is a well-constructed article, with adequate methodology and relevant results and discussion.

It is a very interesting article that portrays the importance of monitoring the work-life quality during COVID-19 pandemic for sustainable management in nursing

Author Response

We thank the editor and all reviewers for their feedback on our manuscript. According to the reviewers' comments, we have prepared the revised manuscript “The importance of monitoring the work-life quality during the COVID-19 restrictions for sustainable management in nursing”. We have revised our manuscript following each raised point and carefully considered all the reviewers' comments. Major corrections have been performed and made English proofreading (added a certificate).

After the review of the English proofreader, we changed the word COVID-19 pandemic to COVID-19 restrictions throughout the entire manuscript - this also led to a minor change in the title.

In the paragraphs below, we list reviewers’ comments, followed by our responses addressed to a specific comment.

All changes in the manuscript we made by Track changes.  

Best regards,

Mateja Lorber, Mojca Dobnik

Reviewer 1:

Comment 1: It is a well-constructed article, with adequate methodology and relevant results and discussion. It is a very interesting article that portrays the importance of monitoring the work-life quality during COVID-19 pandemic for sustainable management in nursing

Response 1: We thank the reviewer for the positive comments.

Round 2

Reviewer 2 Report

The authors have improved the manuscript. Further improvements are possible in the sections they have modified. Revise paying attention to implications that do not seem to emerge at this stage.

Author Response

Dear editor and the reviewer,

We thank the editor and all reviewers for their feedback on our manuscript. According to the reviewers' comments, we have prepared the revised manuscript “The importance of monitoring the work-life quality during the COVID-19 restrictions for sustainable management in nursing”. We have revised our manuscript following each raised point and carefully considered all the reviewers' comments.

In the paragraphs below, we list reviewers’ comments, followed by our responses addressed to a specific comment. All changes in the manuscript we made by Track changes. 

Best regards,

Mateja Lorber, Mojca Dobnik

Reviewer 2:

Comment 1: The authors have improved the manuscript. Further improvements are possible in the sections they have modified. Revise paying attention to implications that do not seem to emerge at this stage.

Response 1: Thank you for your comment. We agree that further improvements are always possible, but we ask the reviewer what he /she suggests to improve in the manuscript. 

We also added a certificate for English proofreading.

Reviewer 3 Report

After reviewing your article again, I found that the variables and hypotheses of this article have been empirically examined in serval studies, so its contribution should not be enough for publication. It is recommended that the author can focus on the theoretical framework and discuss the significant differences from the past literature. Also, you should propose why this research has made a high contribution, because there is no a single theory to fully support your theoretical model, which will represent that it is a data driver article.

Author Response

Dear editor and the reviewer,

We thank the editor and all reviewers for their feedback on our manuscript. According to the reviewers' comments, we have prepared the revised manuscript “The importance of monitoring the work-life quality during the COVID-19 restrictions for sustainable management in nursing”. We have revised our manuscript following each raised point and carefully considered all the reviewers' comments.

In the paragraphs below, we list reviewers’ comments, followed by our responses addressed to a specific comment. All changes in the manuscript we made by Track changes. 

Best regards,

Mateja Lorber, Mojca Dobnik

Reviewer 3:

Comment 1: After reviewing your article again, I found that the variables and hypotheses of this article have been empirically examined in serval studies, so its contribution should not be enough for publication. It is recommended that the author can focus on the theoretical framework and discuss the significant differences from the past literature. Also, you should propose why this research has made a high contribution, because there is no a single theory to fully support your theoretical model, which will represent that it is a data driver article.

Response 1: Thank you for your comments. We add to the Introduction (line 128-145) why we decided on the present research, the gap, the new contribution, and a few sentences into the Discussion (line 361-369).

After reviewing the literature, we can say that during the COVID-19 pandemic, there were almost no studies to examine the work-life quality of employees in nursing. The situation during the COVID-19 pandemic was uncertain and had the greatest impact on the healthcare systems, so the healthcare workers faced daily changes, fear, and uncertainty. To identify research gaps and guide the analysis, the research drew on two literature streams that inform management about professional and private life quality under the COVID-19 restriction for sustainable development. Given the above, we decided to investigate to what extent measures and changes in the functioning of the healthcare system affect the quality of working life to guide management. The present study was broad in scope, it was not only focused on biological characteristics (age, gender, education, working experiences, and marital status) but also on characteristics related to restrictions of the COVID-19 pandemic  (number of patients, accessibility to information, adequate equipment for work, teamwork, use of days off, exposure to stress, managing stress, collaboration), and workplace characteristics (leaders' support, well-being at the workplace), because we want to found out the possible number of factors that influence the work-life quality of employees in nursing.

Overall literature review of work-life quality research showed that work-life quality before the COVID-19 pandemic includes some factors considered personal feelings or perceptions toward work, organization, and employers. In the future, we recommend a qualitative study to understand work-life quality in nursing. The results of these studies can be used as a basis for developing context-based instruments for measuring the work-life quality in nursing. Nerveless, the research has a major contribution to current research and practice for sustainable management and health care not only in the Slovenian area and not only during the COVID-19 restriction period and is therefore a data driver.

We also added a certificate for English proofreading.